# Angular Trajectory of the Vestibular Aqueduct in a Cohort of Chinese Patients with Unilateral Ménière’s Disease: Association with Other Imaging Indices and Clinical Profiles

**DOI:** 10.3390/biomedicines12092008

**Published:** 2024-09-03

**Authors:** Kaijun Xia, Ping Lei, Yingzhao Liu, Cen Chen, Hongjun Xiao, Yangming Leng, Bo Liu

**Affiliations:** 1Department of Otorhinolaryngology-Head and Neck Surgery, Union Hospital, Tongji Medical College, Huazhong University of Science and Technology, Wuhan 430022, China; 2Department of Radiology, Union Hospital, Tongji Medical College, Huazhong University of Science and Technology, Wuhan 430022, China; 3Hubei Province Key Laboratory of Molecular Imaging, Wuhan 430022, China

**Keywords:** Ménière’s disease, high-resolution computer tomography, vestibular aqueduct, endolymphatic sac, angular trajectory of the vestibular aqueduct

## Abstract

Objective: We aimed to investigate the association between the angular trajectory of the vestibular aqueduct (ATVA) with other radiological parameters of temporal bone and clinical characteristics in patients with Ménière’s disease (MD). Methods: A total of 125 unilateral MD patients and 118 controls were enrolled. Computer tomography (CT)-based radiological parameters included ATVA, vestibular aqueduct (VA) visibility, VA morphology, the vertical part of the posterior semicircular canal—the posterior fossa distance (PPD), and peri-VA pneumatization. The clinical characteristics of MD patients included gender, age of diagnosis/onset, disease duration, migraine history, clinical staging, and the results of audio-vestibular tests. The radiological parameters and clinical characteristics in MD patients were compared. Results: Compared with control ears, ATVA ≥ 140° was more prevalent and ATVA ≤ 120° was less frequent in the MD-affected side. For the MD-affected side, MD patients with ATVA ≥ 140° exhibited more severe VA invisibility and obliteration and higher male preponderance than those with ATVA ≤ 120°. Other radio-clinical features did not differ between these two subgroups. Conclusion: In the current study, ATVA ≥ 140°, an indicator of a hypoplastic endolymphatic sac, was found in approximately one-third of the affected and unaffected ears of patients with MD, as well as in a minority of controls. This suggests that the indices may be a predisposing factor rather than a specific marker for the MD ear. The male preponderance in MD patients with hypoplastic ES suggests a gender difference in the anatomical factors for MD pathogenesis.

## 1. Introduction

Ménière’s disease (MD) is an inner ear disease characterized by episodic vertigo, fluctuating hearing loss, tinnitus, and/or ear fullness [1,2]. The etiology and pathogenesis of MD are not fully understood. Endolymphatic hydrops (ELH) is a pathological feature of MD. According to the longitudinal flow theory, endolymph is mainly produced in the stria vascularis and reabsorbed in the endolymphatic sac (ES) and endolymphatic duct (ED) [3]. Anatomical variations of the vestibular aqueduct (VA) and ES may result in ELH by affecting the circulation of endolymph.

Pathological changes of ED/ES, including reduced surface area, narrow width, fibrotic changes in the perisaccular tissue, vascular impairment in the ES etc., may serve as an anatomic basis for MD [4,5,6,7]. Radiological studies have also suggested a role of VA and/or ES variations in the pathogenesis of MD [8]. Mainnemarre et al. detected invisible VA in 42.8% of the MD-affected ear, whereas VA non-visualization was not observed in the normal group [9]. Yamane et al. found that the incidence of obliterated VA in MD patients (64%) was higher than that in the control group (11.5%) [10]. The vertical part of the posterior semicircular canal—the posterior fossa distance (PPD), an indirect indicator of ES size—has been reported to be shorter in MD patients than in normal individuals [11]. Additionally, other anatomical variations, including jugular bulb abnormalities and poor periaqueductal pneumatization, have been observed in patients with MD [12,13].

Two distinct types of ES pathologies, i.e., hypoplastic ES and degenerative ES, have been consistently observed in MD patients, which may correspond to heterogenous clinical features and etiologies [14]. The angular trajectories of the vestibular aqueduct (ATVA) can indirectly reflect the pathological types of ES, thus serving as a surrogate radiological marker in vivo for MD subgrouping. Hypoplastic ES associated with abnormally straight VA may be attributed to developmental factors, resulting in a lack of extraosseous endolymphatic sac (eES) pathologically and ATVA ≥ 140° radiologically, while degenerative ES associated with maturely curved VA may be caused by postnatal factors, leading to presence of eES pathologically and ATVA ≤ 120° radiologically [15]. In addition, gender, frequency of vertigo attacks, and caloric response are reported to differ between MD patients with hypoplastic ES (MD-hp) and those with degenerative ES (MD-dg) [16].

At present, few studies have investigated the role of ATVA in MD [15,16,17,18,19,20,21]. The reliability and clinical effectiveness of ATVA needs further verification. In addition, the relationship between ATVA and other radiological features of VA and ES has not been fully elucidated. The aim of this study is to analyze this relationship in a cohort of Chinese unilateral MD patients and further to explore the association between ATVA and clinical profiles of MD patients.

## 2. Patients and Methods

### 2.1. Participants

This retrospective study reviewed the radiological and clinical data of 125 unilateral definite MD patients who attended the Department of Otorhinolaryngology, Head and Neck Surgery, Union Hospital, Tongji Medical College, Huazhong University of Science and Technology from 2016 to 2023. The diagnosis and staging of MD were established following the diagnostic guidelines proposed by Classification Committee of the Bárány Society in 2015 [1]. The exclusion criteria were as follows: (1) middle- or inner-ear malformation; (2) middle- or inner-ear infections (otitis media, mastoiditis, labyrinthitis, etc.); (3) retro-cochlear lesions (vestibular schwannoma, internal acoustic canal stenosis, etc.); (4) history of ear surgery or intratympanic injections; (5) systemic diseases; and (6) disorders of the central nervous system (vestibular migraine, multiple sclerosis, cerebellar infarction, etc.). The control group included 118 patients with trauma (excluding ears with temporal bone fracture, previous otitis media, vertigo, and hearing loss).

This study was conducted in compliance with the tenets of the Declaration of Helsinki. Informed consent was obtained from each patient and control. The project was approved by the ethical committee of Tongji Medical College of Huazhong University of Science and Technology.

### 2.2. Radiological Evaluation

#### 2.2.1. Computer Tomography (CT) Protocols

CT examinations were conducted using a 64-detector spiral CT scanner (Somatom Definition AS+, Siemens, Forcheim, Germany). All subjects in this study were scanned in the craniocaudal direction while in the supine position. The scan plane was parallel to the orbitomeatal line. The scan parameters were as follows: tube voltage 120 kV, CASE Dose 4D quality reference mAs: 180 mAs, slice thickness 0.6 mm, slice collimation 128 × 0.6 mm, pitch 0.5, field of view 150 mm, reconstruction increment 0.3 mm, and reconstruction kernel H60s.

All CT images were transferred and analyzed on a picture archiving and communication system (PACS) workstation (Carestream Client, Carestream Health). Multi-plane reconstruction in axial, coronal, and sagittal planes was performed. Radiological data were reviewed by two senior neuroradiologist who were blinded to the clinical data.

#### 2.2.2. Radiological Measurement

The visibility of VA in HRCT was graded as previously reported [9]: grade 0, continuous VA; grade I, discontinuous VA; grade II, invisible VA (Figure 1). 

We used Yamane classification criteria to describe the morphology of VA [10]: (A) funnel type, (B) tubular type, (C) filiform type, (D) hollow type, (E) obliterated type (Figure 2). 

Peri-VA pneumatization was categorized according to published classification methods [22]: type 1: large-cell pneumatization in the vicinity of the VA; type 2: small-cell pneumatization in the vicinity of the VA; and type 3: absence of air cells (Figure 3). 

The PPD was measured between the vertical part of the posterior semicircular canal and the posterior fossa on axial panel [11] (Figure 4).

All types of otic capsule dehiscence [23] were evaluated on CT images in at least three different planes, such as axial, coronal, sagittal, Pöschl, and Stenver plane. When a dehiscence was present in at least two consecutive images in all three different planes, it was considered as a true dehiscence (Figure 5).

ATVA was measured as described by Bächinger et al. (http://links.lww.com/MAO/A755, accessed on 14 September 2022) [15] using a specifically developed software (downloaded at https://github.com/DanielZuerrer/CoolAngleCalcJS) (Figure 6). The first image imported to the software showed the horizontal semicircular canal (SCC) and the vestibule. The magenta-shaded area form was adjusted to fit into the boundaries of the vestibule and the horizontal SCC (especially the boundaries of the vestibule), and the first red line was defined. The second imported image showed the distal portion of the VA. A yellow line was adjusted to be parallel with the distal portion of the VA. ATVA was determined between the first red line and the second yellow line. And ears were further divided into ears with ATVA ≤ 120°,120° < ATVA < 140°, and ATVA ≥ 140°.

### 2.3. Audio-Vestibular Examinations

#### 2.3.1. Caloric Test

A caloric test was performed using an infrared video nystagmograph (Visual Eyes VNG, Micromedical Technologies, Chatham, IL, USA). Subjects lay supine with their heads raised 30°, and their ears were alternately filled with warm and cold air (24 °C and 50 °C) for 40 s. The maximum slow phase velocity (SPV_max_) was measured, and the value of canal paresis (CP) was calculated based on the traditional Jongkes formula. A CP value ≥ 25% is considered as an abnormality in the caloric test.

#### 2.3.2. Electrocochleogram (ECochG)

A standard brainstem evoked response device (Nicolet CompassMeridian Systems, Nicolet Biomedical Inc., Madison, WI, USA) was used for electrocochleogram. The silver ball electrode was placed in the deep part of the external auditory canal, and the reference electrode and ground electrode were placed on the ipsilateral earlobe and forehead. Summating potential (SP) and action potential (AP) was recorded in response to click a stimulus (frequency of 11 times per second (95 dB nHL), averaging the responses to 1000 stimuli within 10 ms, and bandpass filtering at 150–3000 Hz). SP/AP ≥ 0.4 was deemed abnormal (positive), indicating the presence of ELH.

#### 2.3.3. Glycerol Test

The subjects were given 50% glycerol orally at a dose of 2.4 mL per kg weight. A pure-tone audiometry test was conducted before and 1, 2, and 3 hours after glycerol intake. The following situations were considered as positive if the hearing threshold was improved by (1) ≥10 dB on at least 3 frequencies and (2) ≥15 dB on one frequency at any time.

### 2.4. Statistical Analysis

The data were analyzed by SPSS 25.0 statistical software. Continuous variables were expressed as mean ± standard deviation or median (quartile). Categorical variables and grade variables were expressed as frequency (percentage). A t-test was used to compare continuous variables of two independent samples with normal distribution. A Mann–Whitney U test was used to compare grade variables or continuous variables of skewness distribution between two independent samples. Categorical variables were compared using Chi square test or Fisher’s exact test for two independent samples. A Shapiro–Wilk test was used to test the normality of data. The inter-rater reliability of the radiological results was determined using intra group correlation coefficient (ICC) for measurement data and kappa value for count data and grade data, respectively. The interpretation of agreement was as follows: ICC ≤ 0.20, poor; 0.2 < ICC ≤ 0.40, fair; 0.4 < ICC ≤ 0.60, moderate; 0.6 < ICC ≤ 0.80, good; 0.8 < ICC ≤ 1.0, excellent. Kappa ≤ 0.20, poor; 0.21<kappa ≤ 0.40, fair; 0.41 < kappa ≤ 0.60, moderate; 0.61 < kappa ≤ 0.80, good; 0.81 < kappa ≤ 1.0, excellent. Good to excellent inter-observer agreement was found in our radiological evaluation. Statistical significance was set as *p* < 0.05. Post hoc analysis was performed if necessary.

## 3. Results

In the present study, 125 unilateral MD patients (125 affected ears and 125 non-affected ears) and 118 control subjects (163 ears) were included. In unilateral MD patients, the age was 52 (38.5, 57) and the male/female ratio was 47/78. In the control group, the age was 48.9 ± 15.4, and the male/female ratio was 40/78.

### 3.1. Distribution of ATVA Subgroups in Control Subjects and MD Patients

As shown in Figure 7, the prevalence of ATVA ≤ 120°, 120° < ATVA < 140°, and ATVA ≥ 140° was 84.7% (138/163), 8.0% (13/163), and 7.4% (12/163) in control ears and 61.6% (77/125), 10.4% (13/125) and 28.0% (35/125) in MD-affected ears, respectively. A statistical difference was detected ($\chi^{2}\;$ = 25.788, *p* < 0.001). Post hoc analysis revealed that compared with control ears, ATVA ≥ 140° was more prevalent ($\chi^{2}$ = 22.065, *p* < 0.001), whereas ATVA ≤ 120° was less frequent ($\chi^{2}$ = 19.886, *p* < 0.001) in MD-affected ears.

Pairwise comparison revealed insignificant interaural difference in the distribution of ATVA subgroups in MD patients ($\chi^{2}$ = 1.856, *p* = 0.603, shown in Figure 7). ATVA ≤ 120° was found in 61.6% (77/125) and 65.6% (82/125) of the affected ears and non-affected ears. It was detected that 120° < ATVA < 140° in 10.4% (13/125) and 12.0% (15/125) of the affected ears and non-affected ears, respectively. And ATVA ≥ 140°was found in 28.0% (35/125) and 22.4% (28/125) of the affected ears and non-affected ears.

### 3.2. The Relationship between ATVA and Other Radiological Variations in Temporal Bone

#### 3.2.1. The Relationship between ATVA and VA Visibility and Morphology

For VA visibility in control ears with ATVA ≥ 140°, grade 0 and grade I were detected in 66.7% (8/12) and 33.3% (4/12) of the ears, respectively, whereas in those with ATVA ≤ 120°, the prevalence of grade 0 and grade I was 90.6% (125/138) and 9.4% (13/138), respectively. A statistical difference was seen in VA visibility between the above subgroups (*Z* = −2.498, *p* = 0.012). For VA visibility in affected ears with ATVA ≥ 140°, the prevalence of grade 0 and grade I were 57.1% (20/35) and 42.9% (15/35), respectively, whereas in those with ATVA ≤ 120°, the prevalence was 80.5% (62/77) and 19.5% (15/77), respectively. MD-affected ears with ATVA ≥ 140° showed reduced VA visibility compared with those with ATVA ≤ 120° (*Z* = −2.578, *p* = 0.01). VA visibility was rated as grade 0 and grade I in 46.4% (13/28) and 53.6% (15/28) of the non-affected ears with ATVA ≥ 140°, respectively, and in 82.9% (68/82) and 17.1% (14/82) of the non-affected ears with ATVA ≤ 120°, respectively. A significant difference was detected (*Z* = −3.767, *p* < 0.001) (Figure 8).

In both control ears and MD-affected ears, the prevalence of VA morphology subtypes differed between the ATVA ≤ 120° subgroup and ATVA ≥ 140° subgroup (for control ears: $\chi^{2}$ = 18.55, *p* = 0.001; for MD-affected ears, $\chi^{2}$ = 12.552, *p* = 0.006), and post hoc analysis revealed higher prevalence of obliterated VA in the ATVA ≥ 140° subgroup (for control ears: $\chi^{2}$ = 5.991, *p* = 0.014; for MD-affected ears: $\chi^{2}$ = 10.429, *p* = 0.001). Type A (funnel), type B (tubular), type C (filiform), type D (hollow) and type E (obliterated) was observed in 16.7% (2/12), 33.3% (4/12), 16.7% (2/12), 8.3% (1/12) and 25.0% (3/12) of the control ears with ATVA ≥ 140°, while the prevalences of type A, type B, type C, type D and type E were 13.0% (18/138), 50.7% (70/138), 30.4% (42/138), 0.0% (0/138) and 5.8% (8/138) in control ears with ATVA ≤ 120°, respectively. Types A, B, C, D, E were observed in 8.6% (3/35), 22.9% (8/35), 28.6% (10/35), 0.0% (0/35), and 40.0% (14/35) of the affected ears with ATVA ≥ 140° and in 22.1% (17/77), 39.0% (30/77), 26.0% (20/77), 0.0% (0/77), and 13.0% (10/77) of the affected ears with ATVA ≤ 120°.

In non-affected ears, the prevalence of types A, B, C, D and E VA was 17.9% (5/28), 7.1% (2/28), 32.1% (9/28), 3.6% (1/28), and 39.3% (11/28) in the ATVA ≥ 140° subgroup, whereas the prevalence was 17.1% (14/82), 43.9% (36/82), 37.8% (31/82), 0.0% (0/82), and 1.2% (1/82) in the ATVA ≤ 120° subgroup. A statistical difference in VA morphology was found between subgroups of non-affected ears ($\chi^{2}$ = 35.503, *p* < 0.001). Compared with the ATVA ≤ 120° subgroup, obliterated VA was more prevalent ($\chi^{2}$ = 12.474, *p* < 0.001) and funnel VA was less frequent ($\chi^{2}$ = 31.120, *p* < 0.001) in the ATVA ≥ 140° subgroup (Figure 8).

#### 3.2.2. The Relationship between ATVA and PPD

As shown in Figure 9, in both control ears and MD-affected ears, PPD was comparable between the ATVA ≥ 140° subgroup and the ATVA ≤ 120° subgroup. For control ears, PPD was 1.94 (1.145, 2.505) mm and 2.105 (1.3675, 3.31) mm in the ATVA ≥ 140° subgroup and the ATVA ≤ 120° subgroup, respectively (*Z* = −0.786, *p* = 0.432). For MD-affected ears, PPD was 1.65 (0.89, 2.71) mm and 1.68 (0.89,2.85) mm in the ATVA ≥ 140° subgroup and the ATVA ≤ 120° subgroup, respectively (*Z* = −0.895, *p* = 0.371). No correlation was found between ATVA and PPD (for control ears: *r* = −0.117, *p* = 0.138; for MD-affected ears: *r* = −0.04, *p* = 0.658), while in non-affected ears, PPD in ears with ATVA ≥ 140° was shorter than those with ATVA ≤ 120° (*Z* = −2.271, *p* = 0.023). The PPD in the former subgroup was 1.145 (0.775, 2.305) mm and was 1.960 (1.128, 2.803) mm in the latter subgroup. A mild negative correlation between ATVA and PPD was observed (*r* = −0.192, *p* = 0.032).

#### 3.2.3. The Relationship between ATVA and Peri-VA Pneumatization

As shown in Figure 10, for peri-VA pneumatization, type 1 (large-cell), 2 (small-cell), and 3 (absence of air cells) were found in 8.3% (1/12), 16.7% (2/12), and 75.0% (9/12) of the control ears with ATVA ≥ 140° and in 31.2% (43/138), 16.7% (23/138), and 52.2% (72/138) of those with ATVA ≤ 120°. The prevalence of type 1, type 2, and type 3 was 8.6% (3/35), 22.9% (8/35), and 68.6% (24/35) in affected ears with ATVA ≥ 140°, whereas the prevalence was 23.4% (18/77), 18.2% (14/77), and 58.4% (45/77) in those with ATVA ≤ 120°. No statistical difference was found (for control ears: *Z* = −1.694, *p* = 0.090; for MD-affected ears, *Z* = −1.357, *p* = 0.175).

In non-affected ears, type 1, 2, and 3 of peri-VA pneumatization were identified in 7.1% (2/28) 0.0% (0/28), and 92.9% (26/28) of the ears with ATVA ≥ 140° and in 24.4% (20/82), 24.4% (20/82), and 51.2% (42/82) of those with ATVA ≤ 120°. Poorer peri-VA pneumatization was found in non-affected ears with ATVA ≥ 140° than in those with ATVA ≤ 120° (*Z* = −3.629, *p* < 0.001).

#### 3.2.4. The Relationship between ATVA and Otic Capsule Dehiscence

As shown in Figure 11, otic capsule dehiscence was found in 15/163 (9.2%) of the control ears, 8/125 (6.4%) of the affected ears, and 12/125 (10.4%) of the non-affected ears. No statistical difference in otic capsule dehiscence was found among the above groups ($\chi^{2}$ = 1.322, *p* = 0.516). Two types of otic capsule dehiscence were found in the present study, i.e., vestibular-aqueduct–jugular-bulb-dehiscence (VA-JBD) and superior semicircular canal dehiscence (SSCD). A combination of VA-JBD and SSCD was detected in one control ear.

The prevalences of VA-JBD in control ears, affected ears, and non-affected ears were 9/163 (5.5%), 4/125 (3.2%), and 6/125 (4.8%), respectively. And the prevalences of SSCD were 7/163 (4.3%), 4/125 (3.2%), and 6/125 (4.8%), respectively. No statistical differences were found in either VA-JBD or SSCD among control ears, affected ears, or non-affected ears (for VA-JBD: $\chi^{2}$ = 0.885, *p* = 0.642; for SSCD: $\chi^{2}$ = 0.427, *p* = 0.808).

In affected ears with ATVA ≤ 120°, 120° < ATVA < 140° and ATVA ≥ 140°, otic capsule dehiscence was detected in 7/77 (9.1%), 1/13 (7.7%), and 0/35 of the ears. No significant difference was detected ($\chi^{2}$ = 3.360, *p* = 0.145).

### 3.3. The Relationship between Clinical Features and ATVA

As shown in Figure 12, the male/female ratio was higher in the MD-140 subgroup (19/16) than the MD-120 subgroup (20/57) ($\chi^{2}$ = 8.498, *p* = 0.004). The migraine history was inquired about in only 67 patients, in which 20% (4/20) of the MD-140 patients and 42.6% (20/47) of the MD-120 patients had a prior history of migraines. No significant differences in migraine history were found between MD-140 subgroup and MD-120 subgroup ($\chi^{2}$ = 3.104, *p* = 0.078).

Age (*Z* = −1.282, *p* = 0.200), course of disease (*Z* = −1.185, *p* = 0.236), age of onset (*Z* = −0.810, *p* = 0.418), and MD stage (*Z* = −0.053, *p* = 0.958) did not differ between MD-140 subgroup and MD-120 subgroup. Stage 1, stage 2, stage 3, and stage 4 were identified in 5.7% (2/35), 25.7% (9/35), 54.3% (19/35), and 14.3% (5/35) of the MD-140 patients and in 11.7% (9/77), 15.6% (12/77), 59.7% (46/77), and 13.0% (10/77) of the MD-120 patients.

The PTA was 50.8 ± 17 dB and 51.7 ± 20.6 in the MD-140 subgroup and the MD-120 subgroup, respectively (t = −0.223, *p* = 0.824). In 25 patients who underwent the glycerol test, positive results were yielded in 57.1% (4/7) of MD-140 patients and 55.6% (10/18) of MD-120 patients ($\chi^{2}$ = 0.005, *p* = 1). Sixty-two patients underwent the ECochG test, and 45% (9/20) of MD-140 patients and 57.1% (24/42) of MD-120 patients showed positive results ($\chi^{2}$ = 0.802, *p* = 0.370). The SP/AP ratio was 0.335 (0.253, 0.95) in the MD-140 subgroup and 0.55 (0.28, 1) in the MD-120 subgroup (*Z* = −0.841, *p* = 0.401). One hundred unilateral MD patients underwent the caloric test. CP value was 26.5% (13%, 45.5%) in the MD-140 subgroup and 40% (18%, 64.25%) in the MD-120 subgroup, which were not statistically different (*Z* = −1.539, *p* = 0.124). Abnormal CP was detected in 60% (18/30) of MD-140 patients and in 65.7% (46/70) of MD-120 patients ($\chi^{2}$ = 0.298, *p* = 0.585) (Figure 13).

## 4. Discussion

### 4.1. ATVA in MD Patients

In the present study, the prevalences of ATVA ≤ 120°, 120° < ATVA < 140°, and ATVA ≥ 140° in MD-affected ears were 61.6%, 10.4%, and 28.0%, respectively. Bächinger et al. reported that the prevalences of ATVA ≤ 120° and ATVA ≥ 140° were 55/72 (76.4%) and 17/72 (23.6%) in MD patients, respectively [16]. Moreover, de Pont et al. detected that the prevalences of ATVA ≤ 120°, 120° < ATVA < 140°, and ATVA ≥ 140° in MD-affected ears were 60.5% (69/114), 25.4% (29/114), and 14.0% (16/114), respectively [21]. The prevalences of ATVA ≥ 140° in our study were consistent with the results of Bächinger et al. [16] but higher than those of de Pont et al. [21]. This discrepancy may be due to different imaging techniques. One of the pathological features of hypoplastic ES is the absence of eES [14]. The operculum, which is the posterior orifice of VA, serves as the boundary marker between eES and the intraosseous portion of the endolymphatic sac (iES). A recent study revealed that in 28.0% (28/100) of MD cases who underwent ES surgery, the operculum was invisible during operation [11], indicating hypoplastic ES in these cases. The prevalence of intraoperatively invisible operculum was consistent with the prevalence of ATVA ≥ 140° in affected ears in our study. 

This study revealed no significant interaural difference in ATVA categories in unilateral MD patients. Currently, few studies have compared ATVA between the affected and non-affected ears of unilateral MD patients. Our findings suggested that prenatal factors affecting ATVA in MD patients may have bilateral effect, and thus, ATVA is not suitable for distinguishing affected and non-affected ears. In MD patients, ATVA ≥ 140 ° may result from prenatal factors [16], which could explain several bilateral ES hypoplasia in our present study.

We found higher prevalence of ATVA ≥ 140° in MD-affected ears than in control ears. Of note, a small proportion of control ears exhibited ATVA ≥ 140° (7.5%), which was not entirely consistent with Bächinger et al. (0%) [15]. However, Jung et al. found that ATVA ≥ 140° was present in 4.4% (13/290) of non-MD patients (patients with superior semicircular canal syndrome, intracochlear schwannoma, otosclerosis, or presbycusis, as well as asymptomatic individuals) [20]. We speculated that this subgroup of normal subjects may have hypoplastic ES pathologically without MD-like symptoms, and long-term prospective observations are needed for the possible future development of MD. Therefore, ATVA ≥ 140° may be a predisposing factor rather than a specific marker for the MD ear.

### 4.2. The Relationship between ATVA and Other Radiological Parameters of VA and ES

#### 4.2.1. The Relationship between ATVA and Other Radiological Parameters of VA

Compared with MD-affected ears with ATVA ≤ 120°, those with ATVA ≥ 140° have lower VA visibility and a higher prevalence of obliterated VA. Radiologically discontinuous VA tends to be classified as the obliterated type. Similar findings have been highlighted by Grosser et al. that the ATVA in ears with invisible VA was 149 ± 24.2°, the ATVA in ears with VA width ≤ 1 mm was 111 ± 18.6°, and the ATVA in ears with VA width ≥ 1 mm was 109 ± 17.4° [24]. However, Huang et al. found the bending angle of VA in MD patients was 117.56 ± 10.35°, which was similar to that in control subjects (119.31 ± 15.09°) [25]. The inconsistency between our results and Huang et al.’s may be due to the different measurement methods. In MD patients with hypoplastic ES, prenatal factors may also influence VA configuration and morphology, probably by halting normal bone growth and remodeling in the opercular region [15]. Otherwise, normal VA development is completed before ES degenerates due to acquired factors, resulting a wider VA and a lower prevalence of obliterated VA.

In the present study, 2D measurement was used to evaluate VA visibility and morphology. Recently, several new methods have been developed. Using 3D reconstruction techniques, Weiss et al. showed a higher intra-class correlation coefficient (ICC) in VA volume measurements compared with conventional VA width measurements (at the midpoint of the vestibule and operculum and at the operculum) [26]. Using a 3D model for angle measurements of the VA, Noyalet et al. found a distinct pattern of angulation between VA and the three SCCs [27]. Therefore, the consistency of different imaging evaluation methods for VA warrants further study.

#### 4.2.2. The Relationship between ATVA and PPD

In MD-affected ears, no significant difference was found in PPD between ears with ATVA ≥ 140° and those with ATVA ≤ 120°, and there was no correlation between ATVA and PPD. These findings seem to contrast with the view that PPD can indirectly reflect the size of ES [11]. Bächinger et al. compared the distance between the most posterior border of the posterior SCC and the posterior temporal bone surface (ΔpSCC-post) between MD-hp and MD-dg and found the ΔpSCC-post of the former subgroup was shorter than that of the latter subgroup [17]. In our study, PPD refers to the perpendicular shortest distance between the vertical part of the posterior SCC and the posterior cranial fossa. The disagreement between our results and those of Bächinger et al. may be attributed to the different measurement methods.

Our results revealed no significant difference in PPD between the control ears with ATVA ≥ 140° and those with ATVA ≤ 120° and no correlation between ATVA and PPD in the control ears. Few studies have analyzed the relationship between ATVA and PPD in normal individuals. Fujita and Sando analyzed 10 temporal bone specimens from individuals aged 4 months to 70 years and found no correlation between age and the bending angle of VA, suggesting that the course of VA does not change after birth [28]. Watzke and Bast analyzed petrous bones from two fetuses and three adults and found the length of ES gradually increased from 5 mm to 15 mm [29]. This temporal difference in the development between VA trajectory and ES size may explain the irrelevance between ATVA and PPD in normal adults. 

Compared to non-affected ears with ATVA ≤ 120°, those with ATVA ≥ 140° manifested a shorter PPD and a weakly negative correlation between PPD and ATVA, which was consistent with our expectations. However, the relationship between PPD and ATVA in non-affected ears is different from that in the affected ears, and the underlying mechanism needs further exploration.

#### 4.2.3. The Relationship between ATVA and Peri-VA Pneumatization

MD- affected ears with ATVA ≥ 140° tended to exhibit poorer peri-VA pneumatization than those ears with ATVA ≤ 120°; however, this difference did not reach statistical significance. However, in non-affected ears, the degree of peri-VA pneumatization was significantly lower in ears with ATVA ≥ 140° than those with ATVA ≤ 120°. The result is not completely consistent with previous studies. Bächinger et al. found that, compared to MD-dg, MD-hp presented smaller volume of mastoid air cells [17]. The disagreement between our results and those of Bächinger et al. may be due to different methods for measuring pneumatization. 

There was no significant difference in the peri-VA pneumatization between the control ears with ATVA ≥ 140° and those ears with ATVA ≤ 120°. Temporal bone studies have shown that the peri-VA pneumatization develops postnatally [30], whereas the development of VA trajectory begins after midterm and is completed before early infancy [28,29]. Based on these results, we speculate that in normal individuals, the development of VA trajectory may not parallel that of the peri-VA pneumatization. This may not hold true for MD patients with hypoplastic ES, in which congenital factors may simultaneously affect peri-VA pneumatization and VA bending.

#### 4.2.4. The Relationship between ATVA and Otic Capsule Dehiscence

In our study, the prevalence of otic capsule dehiscence was comparable in the control ears and affected and non-affected ears. Pathologic third windows were proposed as the underlying mechanism of these dehiscence [31,32]. Not all ears with these variants are accompanied by MD [23], and otic capsule dehiscence should be considered in cases with MD-like symptoms. ELH is not rare in patients with SSCD [33,34]. Lorente-Piera et al. reported that 2/6 subjects with otic capsule dehiscence were diagnosed with MD [35]. Our MD series did not have clinical manifestations of otic capsule dehiscence or the third window of the inner ear, and all patients met the diagnostic criteria for MD.

Although no significant difference was detected in the prevalence of otic capsule dehiscence among affected ears with different ATVA subgroups, almost all otic capsule dehiscence were found in affected ears with ATVA ≤ 120°. Bächinger et al., analyzed magnetic resonance imaging (MRI) images of MD patients and reported that the prevalence of semicircular canal dehiscence in the hypoplastic group (29.4%) was higher than in the degenerative group (3.6%) [16]. Thus, the relationship between ATVA and otic capsule dehiscence and the precise underlying mechanism warrant further study.

### 4.3. The Difference in Clinical Feature between MD-120 Subgroup and MD-140 Subgroup

In this study, compared to the MD-140 subgroup, female preponderance is significantly higher in the MD-120 subgroup. Similarly, Bächinger et al. reported that the proportion of female in the MD-120 subgroup and MD-140 subgroup were 54.5% and 5.9%, respectively [16]. The etiology of MD is multifactorial, including anatomical variations, autoimmune diseases, and migraines. Currently, it is known that the prevalence of autoimmune diseases and migraines in females is higher than in males [36,37], which can partly explain the gender difference in this study, indicating that anatomical factors are more likely to predispose males to developing MD.

The MD-120 subgroup was more likely to have migraine history than the MD-140 subgroup (42.6% and 20.0%); however, this difference was not statistically significant. Bächinger et al. reported that the incidence of migraine history in MD-dg and MD-hp subgroups were 36.4% and 23.5%, respectively, and further comparison between male MD-dg and male MD-hp subgroup still revealed no statistical difference [16]. Our results are partially consistent with the findings of Bächinger et al. [16]. Several studies indicate a bidirectional association between migraine and MD [36]. There are reciprocal neurovascular connections between the vestibular system and the trigeminal nervous system; thus, the dysfunction of the trigeminal nerve may affect cochleovestibular function [38]. In addition, vasospasm in migraines may result in impaired endolymphatic circulation and ultimately develop into ELH [39].

In the present study, there was no significant difference in the age of diagnosis/onset and disease duration between the MD-140 subgroup and the MD-120 subgroup. Bächinger et al. found no significant difference in the age of the onset of first symptoms between MD-hp and MD-dg, but the former subgroup showed an earlier symptom onset [16]. They argue that this phenomenon may be due to prenatal factors in MD-hp, which cause a tendency to manifest earlier in life than MD-dg. The inconsistency between our results and those of Bächinger et al. may be due to the fact that the age of onset in the present study may be different from the time when symptoms first appeared.

We noticed that PTA, clinical stage, ECochG, and the glycerol test results did not differ between the MD-140 subgroup and the MD-120 subgroup, which was consistent with previous findings [16,21]. It is believed that hearing loss in MD is due to the bulging of the basilar membrane caused by ELH [40]. As MD progresses, the degeneration of hair cells and ganglion cells may also result in decreased hearing sensitivity [41]. Our results showed that the severity of cochlear dysfunction may be independent of the pathological types of ES. 

In our MD-120 subgroup, CP value was higher than that in the MD-140 subgroup, although the difference was not statistically significant. Bächinger et al. found severer caloric asymmetry in MD-dg than in MD-hp, which may be due to limited vestibular compensation caused by a more sudden manifestation of ES degeneration in MD-dg cases [16].

### 4.4. Strengths and Limitations

To our knowledge, this was the first study addressing the relationship between ATVA and radiological parameters of temporal bone in a Chinese MD cohort, which not only verified the reliability of ATVA but also helped us gain a deeper understanding of the role of VA in the pathogenies of MD. Our results also validated a male predominance in MD-140, which revealed that pathological type of ES in MD patients may be gender-dependent.

Several limitations exist in the present study. Firstly, the diagnosis of MD in this study was based on clinical manifestations and PTA, while MRI of the inner ear with gadolinium enhancement was not performed to validate ELH. Secondly, a history of migraine and audio-vestibular results in some patients were missing, which may lead to certain bias in data analysis. Furthermore, as we know, familial MD represents one subgroup of MD, whose relationship to anatomical factors warrants thorough investigation [42]. Bächinger et al. have shown that a family history of MD is more common in the hypoplasia subgroup (11.8%) than in the degeneration subgroup (3.6%) [16]. In our MD series, no reliable family history of MD could be obtained, so its relationship with anatomical factors should be explored in a future study.

## 5. Conclusions

In the current study, ATVA ≥ 140°, an indicator of hypoplastic ES, was found in approximately one-third of the affected and unaffected ears of patients with MD, as well as in a minority of controls. This suggests that the indices may be a predisposing factor rather than a specific marker for MD ears. The male preponderance in MD patients with hypoplastic ES suggests a gender difference in the anatomical factors for MD pathogenesis.

## Figures and Tables

**Figure 1 biomedicines-12-02008-f001:**
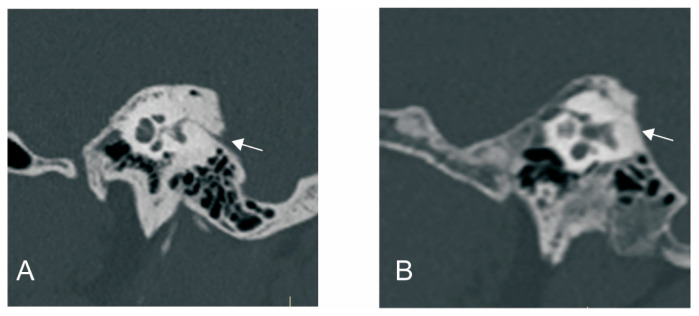
The grading of vestibular aqueduct (VA) visibility on the 45° oblique (Pöschl) planes on temporal bone CT. (**A**) Grade 0 with a continuous VA. (**B**) Grade I with a discontinuous VA. White arrow: VA.

**Figure 2 biomedicines-12-02008-f002:**
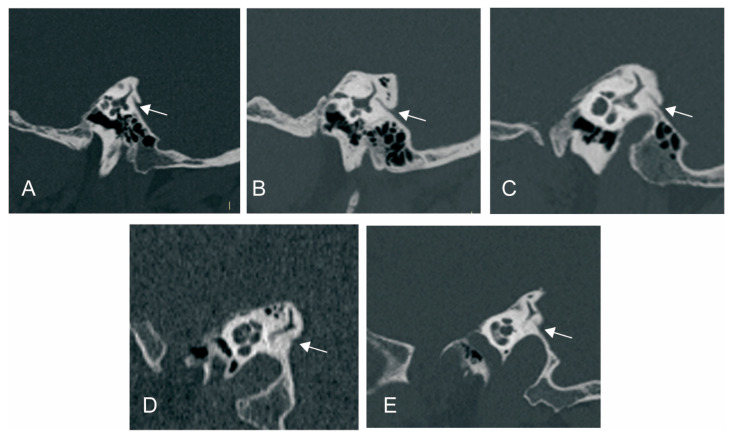
Vestibular aqueduct (VA) morphology subtypes. (**A**) Funnel type; (**B**) tubular type; (**C**) filiform type; (**D**) hollow type; (**E**) obliterated type. White arrow: VA.

**Figure 3 biomedicines-12-02008-f003:**
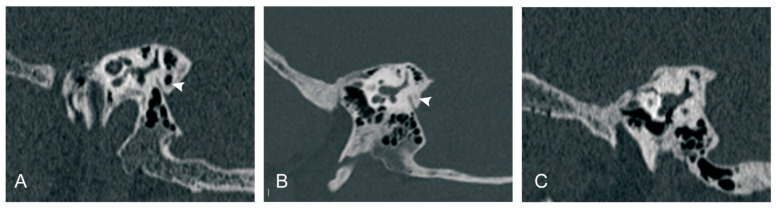
Peri-vestibular aqueduct (VA) pneumatization subtypes. (**A**) Large-cell pneumatization in the vicinity of the VA; (**B**) small-cell pneumatization in the vicinity of the VA; (**C**) absence of air cells. White arrowhead: peri-VA pneumatization.

**Figure 4 biomedicines-12-02008-f004:**
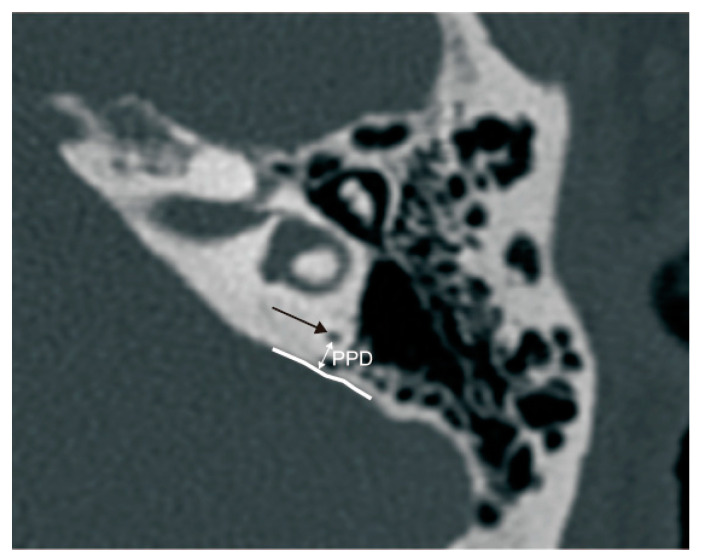
The measurement of the vertical part of the posterior semicircular canal—the posterior fossa distance (PPD). Black arrow: the vertical part of the posterior semicircular canal; white curve: the anterior boundary of the posterior fossa.

**Figure 5 biomedicines-12-02008-f005:**
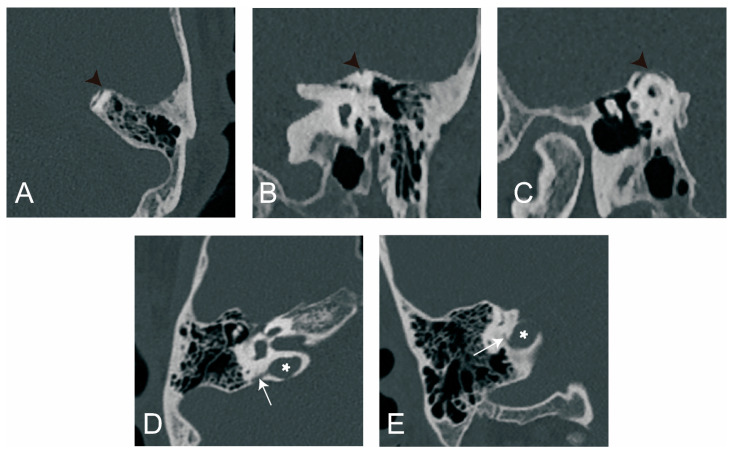
Two types of otic capsule dehiscence. Superior semicircular canal dehiscence (SSCD) on (**A**) axial view, (**B**) coronal view, and (**C**) Pöschl view. Vestibular-aqueduct–jugular-bulb-dehiscence (VA-JBD) on (**D**) the axial view and (**E**) the coronal view. Black arrowhead: SSCD; white arrow: VA; white asterisk: jugular bulb.

**Figure 6 biomedicines-12-02008-f006:**
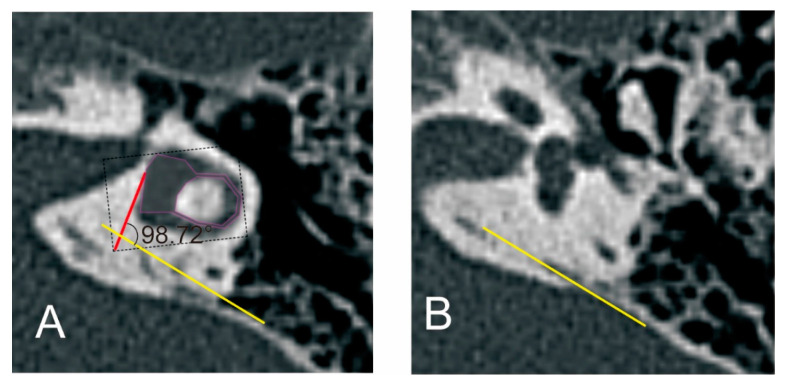
The measurement of the angular trajectory of the vestibular aqueduct (ATVA). (**A**) Images show the horizontal semicircular canal (SCC) and the vestibule. (**B**) Images show the distal portion of the vestibular aqueduct (VA). ATVA refers to the angle between the red line and the yellow line.

**Figure 7 biomedicines-12-02008-f007:**
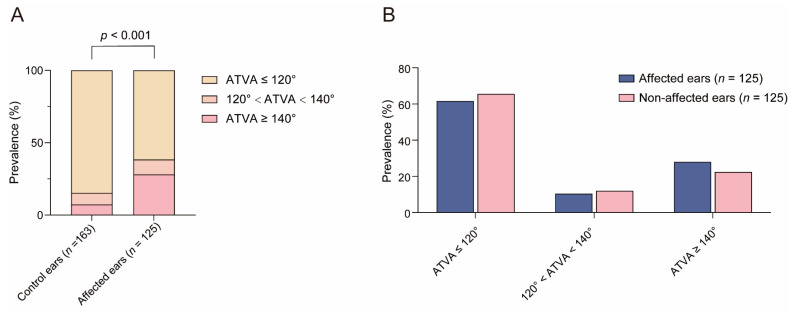
The distribution of angular trajectory of the vestibular aqueduct (ATVA) categories. (**A**) The prevalence of ATVA categories in the affected ears of unilateral Ménière’s disease (MD) patients and control ears. (**B**) Paired comparison of ATVA categories in unilateral MD patients.

**Figure 8 biomedicines-12-02008-f008:**
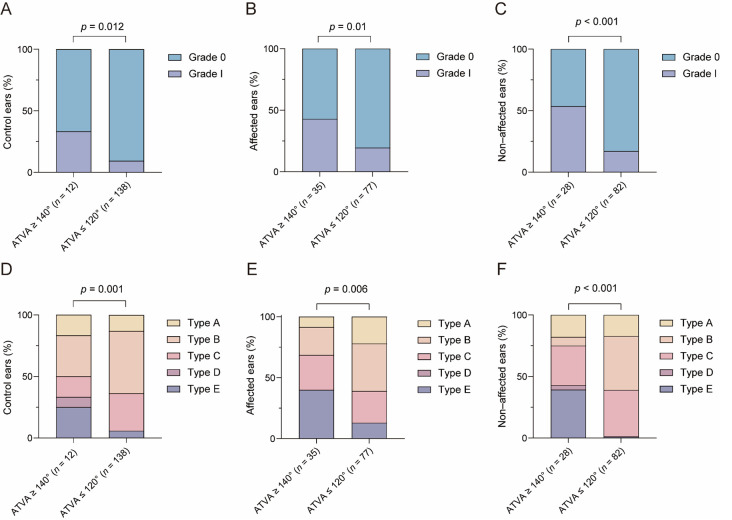
The relationship between angular trajectory of the vestibular aqueduct (ATVA) and vestibular aqueduct (VA) visibility and morphology. The VA visibility in (**A**) control ears, (**B**) affected ears, and (**C**) non-affected ears. The VA morphology in (**D**) control ears, (**E**) affected ears, and (**F**) non-affected ears. Type A, funnel VA; type B, tubular VA; type C, filiform VA; type D, hollow VA; type E, obliterated VA. n, number of ears.

**Figure 9 biomedicines-12-02008-f009:**
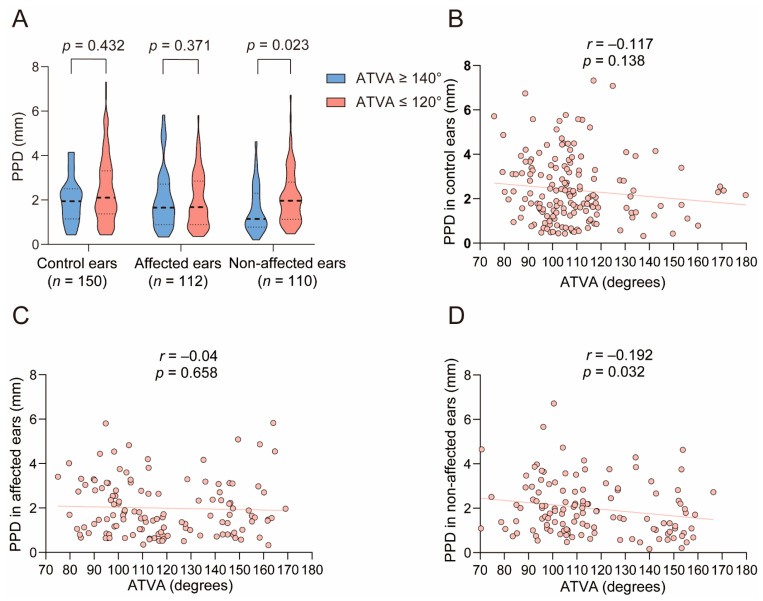
The relationship between angular trajectory of the vestibular aqueduct (ATVA) and vertical part of the posterior semicircular canal—the posterior fossa distance (PPD). (**A**) Comparison of PPD between ears with ATVA ≥ 140° and ears with ATVA ≤ 120° in affected and non-affected ears of unilateral Ménière’s disease (MD) patients and control ears. The correlation between PPD and ATVA in (**B**) control ears, (**C**) affected, and (**D**) non-affected ears of unilateral MD patients.

**Figure 10 biomedicines-12-02008-f010:**
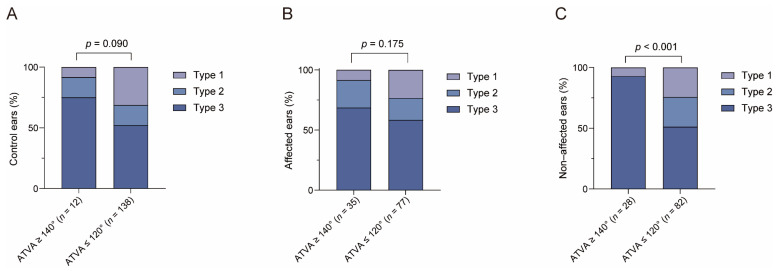
The relationship between angular trajectory of the vestibular aqueduct (ATVA) and peri- vestibular aqueduct (VA) pneumatization. Comparison of peri-VA pneumatization among different ATVA subgroups in (**A**) control ears, (**B**) affected ears, and (**C**) non-affected ears. n, number of ears.

**Figure 11 biomedicines-12-02008-f011:**
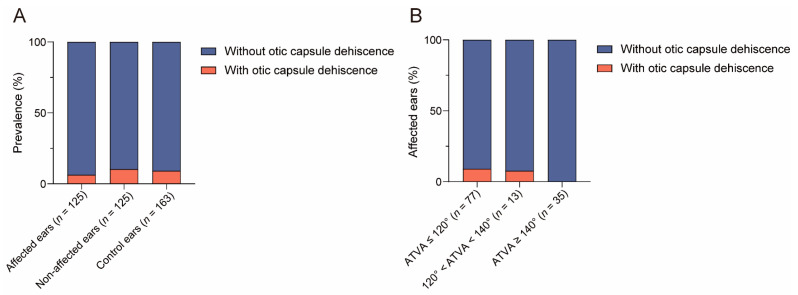
The relationship between angular trajectory of the vestibular aqueduct (ATVA) and otic capsule dehiscence. (**A**) The prevalence of otic capsule dehiscence in affected ears, non-affected ears, and control ears. (**B**) The prevalence of otic capsule dehiscence in affected ears with different ATVA subgroups. n, number of ears.

**Figure 12 biomedicines-12-02008-f012:**
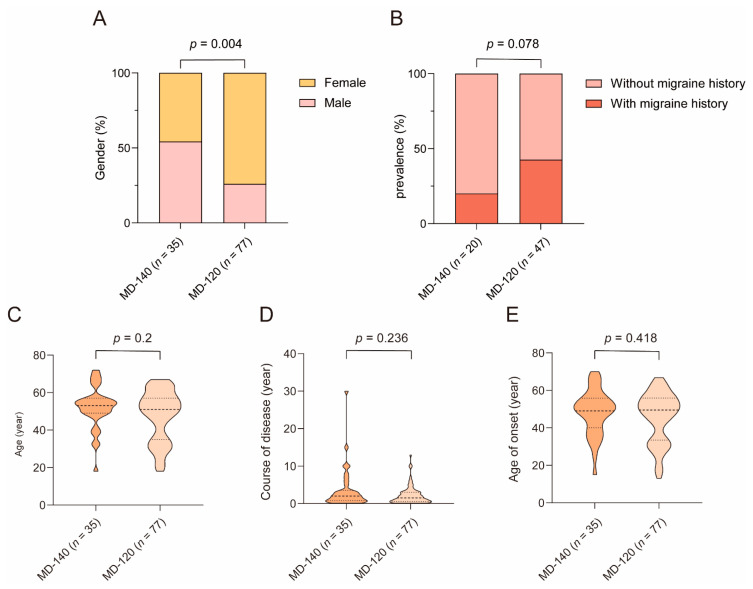
The clinical features of MD-140 and MD-120. (**A**) Gender, (**B**) migraine history, (**C**) age, (**D**) course of disease, and (**E**) age of onset were compared between MD-140 and MD-120. Ménière’s disease, MD.

**Figure 13 biomedicines-12-02008-f013:**
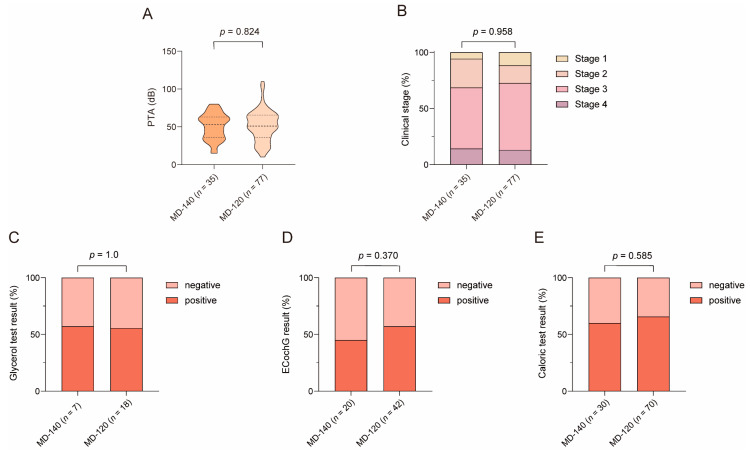
Comparison of cochlear and vestibular functions between MD-140 and MD-120. (**A**) pure-tone average (PTA), (**B**) clinical stage, (**C**) glycerol test result, (**D**) Electrocochleogram (ECochG) result, and (**E**) caloric test result. Ménière’s disease, MD.

## Data Availability

The original contributions presented in the study are included in the article; further inquiries can be directed to the corresponding authors.

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
