# Peer review of "Angular Trajectory of the Vestibular Aqueduct in a Cohort of Chinese Patients with Unilateral Ménière’s Disease: Association with Other Imaging Indices and Clinical Profiles"

_biomedicines, 2024, doi:10.3390/biomedicines12092008_

Round 1
Reviewer 1 Report
Comments and Suggestions for Authors
I think it's a very interesting article. I only ask you small questions:Patients and method: I would like to know why you used trauma patients as a control group. Are you refering to head trauma?. Among the criteria for exclusion from the controls, perhaps a history of vertigo should be included ?. Could you explain why of the 118 controls you only use 163 ears for the study and not the total 236?
I consider that the sections on results and discussion are correct. Except for a comment regarding the study by De Pont, where he finds a LOWER percentage of ATVA>140 in patients with Meniere, compared to the one obtained by you.
Author Response
Comments 1: I think it's a very interesting article. I only ask you small questions: Patients and method: I would like to know why you used trauma patients as a control group. Are you refering to head trauma? Among the criteria for exclusion from the controls, perhaps a history of vertigo should be included? Could you explain why of the 118 controls you only use 163 ears for the study and not the total 236?
Response 1: Thanks for your valuable suggestions and comments.
(1) Clinically, patients usually underwent temporal bone CT scan due to ear symptoms or head trauma. Choosing head trauma patients as controls helped us rule out those subjects with ear symptoms and ensured a sufficient number of control cases.
(2) In the present study, exclusion criteria of controls included a history of vertigo. We feel sorry for the omitted writing.
(3) Seventy-three ears of 118 controls were excluded because of temporal bone fracture or underlying ear diseases:(1) middle or inner ear malformation; (2) middle or inner ear infections (otitis media, mastoiditis, labyrinthitis etc.); (3) retro-cochlear lesions (vestibular schwannoma, internal acoustic canal stenosis etc.); (4) history of ear surgery or intratympanic injections. And 163 ears were included as controls.
Comments 2: I consider that the sections on results and discussion are correct. Except for a comment regarding the study by De Pont, where he finds a LOWER percentage of ATVA>140 in patients with Meniere, compared to the one obtained by you.
Response 2: Thanks for the reviewer's comment regarding the incidence of ATVA>140 in MD patients between previous study and ours.
Our results showed a prevalence of 28% for ATVA ≥140° in MD-affected ears, whereas Bächinger et al. reported a prevalence of 23.6% and de Pont et al. reported 14%. The differences may be due to different imaging evaluation methods. In de Pont et al.'s study, ATVA measurements were performed based on 4h-delayed 3D FLAIR MRI, whereas we and Bächinger et al. used CT measurement. Our recent studies found that the visualization of MRI-VA was significantly poorer than that of CT-VA. A subset of patients with MD had no discernable VA in MRI evaluation, thus ATVA evaluation based on MRI could not be applied in these patients. However, if these patients were evaluated by CT, it is very likely that their ATVA would fall into the category of ≥140°. This may explain the lower prevalence of ATVA ≥140° in the de Pont et al. study than our results and those of Bächinger et al.
In our revised manuscript,Line 336-339 revised as The prevalence of ATVA ≥ 140° in our study were consistent with the results of Bächinger et al., but higher than those of de Pont et al. This discrepancy may be due to different imaging techniques.
References:
[1] Xia K, Lei P, Liu Y, Chen C, Pan H, Leng Y, Liu B. Comparison of vestibular aqueduct visualization on computed tomography and magnetic resonance imaging in patients with Ménière's disease. BMC Med Imaging. 2024; 24(1): 93. doi: 10.1186/s12880-024-01275-8.
[2] Bächinger D, Brühlmann C, Honegger T, Michalopoulou E, Monge Naldi A, Wettstein VG, Muff S, Schuknecht B, Eckhard AH. Endotype-phenotype patterns in Meniere's disease based on gadolinium-enhanced MRI of the vestibular aqueduct. Front Neurol. 2019, 10, 303, doi:10.3389/fneur.2019.00303.
[3] de Pont LMH, Houben MTPM, Verhagen TO, Verbist BM, van Buchem MA, Bommeljé CC, Blom HM, Hammer S. Visualization and clinical relevance of the endolymphatic duct and sac in Meniere's disease. Front Neurol. 2023; 14: 1239422. doi: 10.3389/fneur.2023.1239422.
Reviewer 2 Report
Comments and Suggestions for Authors
This is a case control study comparing the angular trajectory of the VA (ATVA)in 125 Chinese patients with MD and 118 controls. This should be stated since ethnicity may be a critical factor in a developmental condition.
I suggest to change the title and include Chinese and unilateral for clarity.
The study is well-designed, and the results are highly relevant for a better understanding of the physiopathology of MD. The authors describe different VA morphologies according to a previously reported classification as grade 0, I and II and Yamane classification criteria to describe the morphology. They also measured the posterior canal to posterior fossa distance (PPD). A large ATVA and a shorted PPD have been reported to be more common in MD.
I recommend its publication with some minor changes that the authors could address.
In practice, they are revisiting 2 endophenotypes according to temporal bone morphology: hypoplastic endolymphatic sac and non-hypoplastic sac MD patients.
There are some relevant findings in the study:
1. The ATVA was different between unilateral MD and controls, but 7.4% of controls showed ATVA > 140. This is important since the prevalence of MD is <0.001 in the general population and the finding of ATVA >140 is 10 times more common than MD, showing that this finding itself cannot explain the association between endolymphatic sac hypoplasia and MD and additional factors are needed to develop the complete clinical phenotype.
2. The ATVA was not different between affected and non-affected ears in individuals with MD, so we cannot associate ATVA with unilateral MD (Figure 6B).
3. MD associated with migraine is also known as MD type 4 in the clinical classification of Frejo et al (2017). They found that migraine is not associated with ATVA>140, so we can not associate this finding with the MD type 4 clinical variant.
4. No correlation was found in PPD or peri-vascular pneumatization and ATVA (Figures 8 and 9).
I would suggest that the authors should report the familial history of MD in this cohort and compare ATVA and PPD between familial and non-familial cases (MD type 3). Familial MD is found in 9% of European and 6% of East Asian. Several genes have been involved in familial MD, and this should be mentioned in the discussion, since this clinical subgroup can show higher frequency of MD>140.
References regarding the clinical subgroups and genetics of familial MD should be also included in the reference list.
Conclusions should be reformulated to state that 30% of MD and 7% of controls show hypoplastic ES. Moreover, the presence of MD>140 in affected and unaffected ears in the same TC scan and the relatively high frequency in control of ATVA>140 support that this is not a specific finding of MD, given the high inter-observed agreement.
Author Response
Comments 1: This is a case control study comparing the angular trajectory of the VA (ATVA) in 125 Chinese patients with MD and 118 controls. This should be stated since ethnicity may be a critical factor in a developmental condition.
I suggest to change the title and include Chinese and unilateral for clarity.
Response 1: We appreciate the reviewer's valuable feedback and suggestion regarding the clarification of title. In the revised version of the manuscript, the title was revised as: Angular trajectory of the vestibular aqueduct in a cohort of Chinese patients with unilateral Meniere's disease: association with other imaging indices and clinical profiles.
Comments 2: The study is well-designed, and the results are highly relevant for a better understanding of the physiopathology of MD. The authors describe different VA morphologies according to a previously reported classification as grade 0, I and II and Yamane classification criteria to describe the morphology. They also measured the posterior canal to posterior fossa distance (PPD). A large ATVA and a shorted PPD have been reported to be more common in MD.
I recommend its publication with some minor changes that the authors could address.
In practice, they are revisiting 2 endophenotypes according to temporal bone morphology: hypoplastic endolymphatic sac and non-hypoplastic sac MD patients.
There are some relevant findings in the study:
1. The ATVA was different between unilateral MD and controls, but 7.4% of controls showed ATVA > 140. This is important since the prevalence of MD is <0.001 in the general population and the finding of ATVA >140 is 10 times more common than MD, showing that this finding itself cannot explain the association between endolymphatic sac hypoplasia and MD and additional factors are needed to develop the complete clinical phenotype.
2. The ATVA was not different between affected and non-affected ears in individuals with MD, so we cannot associate ATVA with unilateral MD (Figure 6B).
3. MD associated with migraine is also known as MD type 4 in the clinical classification of Frejo et al (2017). They found that migraine is not associated with ATVA>140, so we can not associate this finding with the MD type 4 clinical variant.
4. No correlation was found in PPD or peri-vascular pneumatization and ATVA (Figures 8 and 9).
I would suggest that the authors should report the familial history of MD in this cohort and compare ATVA and PPD between familial and non-familial cases (MD type 3). Familial MD is found in 9% of European and 6% of East Asian. Several genes have been involved in familial MD, and this should be mentioned in the discussion, since this clinical subgroup can show higher frequency of MD>140.
References regarding the clinical subgroups and genetics of familial MD should be also included in the reference list.
Response 2: Thank you for your affirmative and professional comments. We would like to express our gratitude for your valuable feedback on our manuscript.
All patients with vestibular complaints who attended our department were interviewed about past history of autoimmune diseases, migraine and family history. Familial MD has been proposed as a subtype of MD. Although some of our patients reported a family member with Meniere's disease, in most cases the diagnoses were not made by specialists. As we know, the diagnosis of MD requires a detailed history taking and documented audiologic results, and imaging and other audio-vestibular examinations are essential for differential diagnosis. Therefore, it is reasonable to assume that the self-reported familial type of MD in these patients is of low credibility and was not adopted by this study.
Therefore, in the limitations section of the revised manuscript, we add the following. Furthermore, as we know, familial MD represents one subgroup of MD, whose relationship to anatomical factors warrants thorough investigation. Bächinger et al. have shown that a family history of MD is more common in the hypoplasia subgroup (11.8%) than in the degeneration subgroup (3.6%). In our MD series, no reliable family history of MD could be obtained, so its relationship with anatomical factors should be explored in a future study.
Additionally, In the revised version of the manuscript, the references regarding the clinical subgroups and genetics of familial MD were included as following.
References
[1] Frejo L, Martin-Sanz E, Teggi R, Trinidad G, Soto-Varela A, Santos-Perez S, et al. Extended phenotype and clinical subgroups in unilateral Meniere disease: A cross-sectional study with cluster analysis. Clin Otolaryngol. 2017; 42(6): 1172-1180. doi: 10.1111/coa.12844.
[2] Frejo L, Soto-Varela A, Santos-Perez S, Aran I, Batuecas-Caletrio A, Perez-Guillen V, et al. Clinical Subgroups in Bilateral Meniere Disease. Front Neurol. 2016;7:182. doi: 10.3389/fneur.2016.00182.
[3] Bächinger D, Brühlmann C, Honegger T, Michalopoulou E, Monge Naldi A, Wettstein VG, et al. Endotype-Phenotype Patterns in Meniere's Disease Based on Gadolinium-Enhanced MRI of the Vestibular Aqueduct. Front Neurol. 2019; 10: 303. doi: 10.3389/fneur.2019.00303.
Comments 3: Conclusions should be reformulated to state that 30% of MD and 7% of controls show hypoplastic ES. Moreover, the presence of MD>140 in affected and unaffected ears in the same TC scan and the relatively high frequency in control of ATVA>140 support that this is not a specific finding of MD, given the high inter-observed agreement.
Response 3: Thank you for your affirmative and professional comments. We would like to express our gratitude for your valuable feedback on the conclusions section of our manuscript.
The previous conclusions have been reformulated as follow:
In the current study, ATVA ≥ 140 °, an indicator of hypoplastic ES, was found in approximately one third of the affected and unaffected ears of patients with MD, as well as in a minority of controls. This suggests that the indices may be a predisposing factor rather than a specific marker for the MD ear. The male preponderance in MD patients with hypoplastic ES suggests a gender difference in the anatomical factors for MD pathogenesis.
Reviewer 3 Report
Comments and Suggestions for Authors
1. Among the reported local anatomical conditions that can be implicated in the genesis of Menière-type primary hydrpsoa is the high jugular bulb. I suggest the authors specify this in the introduction - line 50/51
2. A big concern related to this study is related to the fact that the authors did not analyze - and did not specify this in the method - if they controlled in the defined MD group not be concurrent with some new or more classic variants of Otic Capsule Dehiscences e.g. SSCD, CochleoFacial or VA/IJV or even beetwen the Horziontal SSC and the second segment of the Facial Nerve. It becomes more and more obvious with the new publications on this topic in the recent literature, that Third Mobile Windows Abnormalities can generate a pathology easily confused with MD. Thus, the authors have the duty to review the manuscript from this point of view, since the conclusions can be erroneous if this source of bias is not avoided
Author Response
Comments 1: Among the reported local anatomical conditions that can be implicated in the genesis of Menière-type primary hydrpsoa is the high jugular bulb. I suggest the authors specify this in the introduction - line 50/51
Response 1: Thank you for your affirmative and professional comments.
In our revised manuscript,Line 51, we added the following sentence.
Additionally, other anatomical variations, including the jugular bulb abnormalities, poor periaqueductal pneumatization, etc, have been observed in patients with MD.
References
[1] Redfern RE, Brown M, Benson AG. High jugular bulb in a cohort of patients with definite Meniere’s disease. J Laryngol Otol. (2014) 128:759-764.
[2] Hall SF, O’Connor AF, Thakkar CH, Wylie IG, Morrison AW. Significance of tomography in Meniere’s disease: periaqueductal pneumatization. Laryngoscope. (1983) 93:1551–1553.
Comments 2: A big concern related to this study is related to the fact that the authors did not analyze - and did not specify this in the method - if they controlled in the defined MD group not be concurrent with some new or more classic variants of Otic Capsule Dehiscences e.g. SSCD, CochleoFacial or VA/IJV or even beetwen the Horziontal SSC and the second segment of the Facial Nerve. It becomes more and more obvious with the new publications on this topic in the recent literature, that Third Mobile Windows Abnormalities can generate a pathology easily confused with MD. Thus, the authors have the duty to review the manuscript from this point of view, since the conclusions can be erroneous if this source of bias is not avoided.
Response 2: Thank you for your professional comments and suggestions.
We have reviewed and re-evaluated the radiological data of our series focusing on otic capsule dehiscences. In the revised version of our manuscript, evaluation method, typical images and statistical analysis have been supplemented in the Method, Results and Discussion section.
In Method section of our revised manuscript, the following sentences have been added:
All types of otic capsule dehiscence were evaluated on CT images in at least three different planes, such as axial, coronal, sagittal, Pöschl and Stenver plane. When a dehiscence was present in at least two consecutive images in all three different planes, it was considered as a true dehiscence (Figure 5).

Figure 5. Two types of otic capsule dehiscence. (A) superior semicircular canal dehiscence (SSCD) on coronal view. (B) vestibular aqueduct-jugular bulb dehiscence (VA-JBD) on axial view. Black arrowhead: SSCD; white arrow: VA; white asterisk: jugular bulb.
In Results section of our revised manuscript, the following sentences have been added:

Figure 11. the relationship between ATVA and otic capsule dehiscence. (A) the prevalence of otic capsule dehiscence in affected ears, non-affected ears and control ears. (B) the prevalence of otic capsule dehiscence in affected ears with different ATVA subgroups. n, number of ears.
As shown in figure 11, otic capsule dehiscence was found in 15/163(9.2%) of the control ears, 8/125(6.4%) of the affected ears and 12/125(10.4%) of the non-affected ears. No statistical difference in otic capsule dehiscence was found among the above groups ( x2=1.322, p=0.516). Two types of otic capsule dehiscence were found in the present study, i.e., VA-JBD and SSCD. A combination of VA-JBD and SSCD was detected in one control ear.
The prevalence of VA-JBD in control ears, affected ears and non-affected ears were 9/163(5.5%), 4/125(3.2%) and 6/125(4.8%), respectively. And the prevalence of SSCD were 7/163(4.3%), 4/125(3.2%) and 6/125(4.8%), respectively. No statistical difference was found in either VA-JBD or SSCD among control ears, affected ears and non-affected ears (for VA-JBD: x2=0.885, p=0.642; for SSCD: x2=0.427, p=0.808).
In affected ears with ATVA≤120°, 120°<ATVA<140° and ATVA≥140°, otic capsule dehiscence was detected in 7/77(9.1%), 1/13(7.7%) and 0/35 of the ears. No significant difference was detected ( x2=3.360, p=0.145).
Otic capsule dehiscence are relatively new and unfamiliar entities and have gained significant attention in recent years. Several variants have been described, including superior, lateral, and posterior semicircular canal dehiscence; carotid-cochlear, facial-cochlear, and internal auditory canal-cochlear dehiscence, etc. The underlying mechanism may be associated with third mobile window in the inner ear. Their clinical manifestations are easily confused with those of MD or other inner ear disorders. Therefore, otic capsule dehiscences should be considered in cases clinically suggestive of MD. Lorente-Piera et al., reported that 2/6 subjects with otic capsule dehiscence were diagnosed with MD. Johanis et al. and Sone et al. also observed ELH as a comorbid condition in SSCD patients. In our series, all MD patients fully met the diagnostic criteria proposed by Barany Society without any signs suggestive of otic capsule dehiscence, such as sound or pressure induced vertigo or dizziness. We found no significant difference in the prevalence of otic capsule dehiscence among MD affected ears, non-affected ears and control ears. The prevalence of otic capsule dehiscence did not differ among ATVA subgroups, neither. Our findings were discrepant from those of Bächinger et al., who reported higher prevalence of semicircular canal dehiscence in hypoplastic group (29.4%) than in degenerative group (3.6%). Therefore, the relationship between ATVA and otic capsule dehiscence warrants further study.
In Discussion section of our revised manuscript, the following sentences have been added:
In our study, the prevalence of otic capsule dehiscence was comparable in the control ears, affected and non-affected ears. Pathologic third windows were proposed as the underlying mechanism of these dehiscence. Not all ears with these variants are accompanied by MD, otic capsule dehiscence should be considered in cases with MD-like symptoms. ELH is not rare in patients with SSCD. Lorente-Piera et al., reported that 2/6 subjects with otic capsule dehiscence were diagnosed with MD . Our MD series did not have clinical manifestations of otic capsule dehiscence or third window of the inner ear, and all patients met the diagnostic criteria for MD.
Although no significant difference was detected in the prevalence of otic capsule dehiscence among affected ears with different ATVA subgroups, almost all otic capsule dehiscence were found in affected ears with ATVA ≤ 120 °. Bächinger et al., analyzed MRI images of MD patients and reported that the prevalence of semicircular canal dehiscence in hypoplastic group (29.4%) was higher than in degenerative group (3.6%). Thus, the relationship between ATVA and otic capsule dehiscence and the precise underlying mechanism warrants further study.
Reference
[1] Ho, M.L.; Moonis, G.; Halpin, C.F.; Curtin, H.D. Spectrum of Third Window Abnormalities: Semicircular Canal Dehiscence and Beyond. AJNR Am. J. Neuroradiol. 2017, 38, 2-9, doi:10.3174/ajnr.A4922.
[2] Lorente-Piera, J.; Prieto-Matos, C.; Manrique-Huarte, R.; Garaycochea, O.; Domínguez, P.; Manrique, M. Otic Capsule Dehiscences Simulating Other Inner Ear Diseases: Characterization, Clinical Profile, and Follow-Up-Is Ménière's Disease the Sole Cause of Vertigo and Fluctuating Hearing Loss? Audiol Res 2024, 14, 372-385, doi:10.3390/audiolres14020032.
[3] Johanis, M.; De Jong, R.; Miao, T.; Hwang, L.; Lum, M.; Kaur, T.; Willis, S.; Arsenault, J.J.; Duong, C.; Yang, I., et al. Concurrent superior semicircular canal dehiscence and endolymphatic hydrops: A novel case series. J. Surg. Case Rep. 2021, 78, 382-386, doi:10.1016/j.ijscr.2020.12.074.
[4] Motasaddi Zarandy, M.; Kouhi, A.; Emami, H.; Amirzargar, B.; Kazemi, M.A. Prevalence of otic capsule dehiscence in temporal bone computed tomography scan. Arch. Otorhinolaryngol. 2023, 280, 125-130, doi:10.1007/s00405-022-07464-x.
[5] Bächinger, D.; Brühlmann, C.; Honegger, T.; Michalopoulou, E.; Monge Naldi, A.; Wettstein, V.G.; Muff, S.; Schuknecht, B.; Eckhard, A.H. Endotype-phenotype patterns in Meniere's disease based on gadolinium-enhanced MRI of the vestibular aqueduct. Neurol. 2019, 10, 303, doi:10.3389/fneur.2019.00303.
Round 2
Reviewer 3 Report
Comments and Suggestions for Authors In Figure 5 B where the authors produced a vascular-labyrinthine OCD type between VA and IJV, the position of the vein is somewhat bizarre - possible IJV trajectory abnormality very close to the Internal Auditory Canal. Please enter, search for a "classic" image for this variant of OCD . Also, provide for each of these two OCD's variants you chose to demonstate in axial and coronal plane - even better in Poschl plane for the SSCD variant.
Author Response
Comments 1: In Figure 5 B where the authors produced a vascular-labyrinthine OCD type between VA and IJV, the position of the vein is somewhat bizarre - possible IJV trajectory abnormality very close to the Internal Auditory Canal. Please enter, search for a "classic" image for this variant of OCD . Also, provide for each of these two OCD's variants you chose to demonstate in axial and coronal plane - even better in Poschl plane for the SSCD variant.
Response 1: Thanks for your valuable suggestions. In the revised version of the manuscript, a more classic image for VA-JBD was presented in Figure 5, and more than two planes were chosen to demonstrate SSCD and VA-JBD. The radiological images and legend of Figure 5 have been revised:
Figure 5. Two types of otic capsule dehiscence. Superior semicircular canal dehiscence (SSCD) on (A)axial view, (B)coronal view and (C)Pöschl view. Vestibular aqueduct-jugular bulb dehiscence (VA-JBD) on (D)axial view and (E)coronal view. Black arrowhead: SSCD; white arrow: VA; white asterisk: jugular bulb.